# Which National Factors Are Most Influential in the Spread of COVID-19?

**DOI:** 10.3390/ijerph18147592

**Published:** 2021-07-16

**Authors:** Hakyong Kim, Catherine Apio, Yeonghyeon Ko, Kyulhee Han, Taewan Goo, Gyujin Heo, Taehyun Kim, Hye Won Chung, Doeun Lee, Jisun Lim, Taesung Park

**Affiliations:** 1Department of Industrial Engineering, Seoul National University, Seoul 08826, Korea; gkrdyd111@snu.ac.kr; 2Interdisciplinary Program in Bioinformatics, Seoul National University, Seoul 08826, Korea; 2019-20240@snu.ac.kr (C.A.); hgh1031@snu.ac.kr (K.H.); gootec92@snu.ac.kr (T.G.); hgj0106@snu.ac.kr (G.H.); 2corin417@snu.ac.kr (D.L.); 3Department of Statistics, Seoul National University, Seoul 08826, Korea; newstellar@snu.ac.kr (Y.K.); qeeqee10@snu.ac.kr (T.K.); 4Department of Archeology and Art History, Seoul National University, Seoul 08826, Korea; 5Department of Chemistry, Seoul National University, Seoul 08826, Korea; hyewon.chung@snu.ac.kr; 6The Research Institute of Basic Sciences, Seoul National University, Seoul 08826, Korea; swanjslim@gmail.com

**Keywords:** COVID-19, SARS-CoV-2, growth curve models, pandemic

## Abstract

The outbreak of the novel COVID-19, declared a global pandemic by WHO, is the most serious public health threat seen in terms of respiratory viruses since the 1918 H1N1 influenza pandemic. It is surprising that the total number of COVID-19 confirmed cases and the number of deaths has varied greatly across countries. Such great variations are caused by age population, health conditions, travel, economy, and environmental factors. Here, we investigated which national factors (life expectancy, aging index, human development index, percentage of malnourished people in the population, extreme poverty, economic ability, health policy, population, age distributions, etc.) influenced the spread of COVID-19 through systematic statistical analysis. First, we employed segmented growth curve models (GCMs) to model the cumulative confirmed cases for 134 countries from 1 January to 31 August 2020 (logistic and Gompertz). Thus, each country’s COVID-19 spread pattern was summarized into three growth-curve model parameters. Secondly, we investigated the relationship of selected 31 national factors (from KOSIS and *Our World in Data*) to these GCM parameters. Our analysis showed that with time, the parameters were influenced by different factors; for example, the parameter related to the maximum number of predicted cumulative confirmed cases was greatly influenced by the total population size, as expected. The other parameter related to the rate of spread of COVID-19 was influenced by aging index, cardiovascular death rate, extreme poverty, median age, percentage of population aged 65 or 70 and older, and so forth. We hope that with their consideration of a country’s resources and population dynamics that our results will help in making informed decisions with the most impact against similar infectious diseases.

## 1. Introduction

The novel coronavirus disease 2019 (COVID-19), a highly transferable viral disease, is a respiratory illness caused by novel severe acute respiratory syndrome coronavirus 2 (SARS-CoV-2), which has person-to-person contact as the main route of transmission and causes flu-like symptoms and in severe cases death [1,2]. The spread of COVID-19 became a global threat, and the World Health Organization (WHO) declared it a global pandemic on March 11, 2020 [3]. The public health threat it represents is the most severe that has been seen in respiratory viruses since the 1918 H1N1 influenza pandemic [4], with a total of 104,904,529 confirmed cases and 2,278,471 deaths worldwide, as of 4 February 2021 [5].

Many factors can influence the epidemiological characteristics and contribute to the increased mortality rate of COVID-19 [6,7,8,9,10]. Therefore, understanding the potential affecting factors involved in the outspread of COVID-19 will be of great significance in containing the spread of the epidemic and was the focus of many studies [6]. Several theories are suggesting the impact of environmental factors such as temperature and humidity, along with international travel and lack of proper social consciousness for isolation as causes of the global spread of COVID-19 [1]. A few investigations have also considered social aspects potentially associated with the spread of COVID-19, such as population density, metropolitan population, intra-provincial traffic, and national lockdown, indicating that the social distancing measures have been successful in reducing new cases [7,8,9,10].

Furthermore, several studies have investigated the impact of weather on the COVID-19 transmission, with special attention being paid to temperature and humidity, indicating that temperature is inversely related to COVID-19 incidence [11]. Moreover, each 1 °C increase in temperature was found to be associated with decreases in daily new cases at different extents. Significant studies on the impacts of climatic predictors on COVID-19 transmission were also conducted in China, the United States, and Europe [12,13].

A look at history tells us that pandemics and epidemics have consistently and significantly affected human lives, and that governments have continually tried to find ways of slowing down the spread of these diseases; for example, quarantines were employed during the Ebola outbreak in West Africa [14,15]. The objective of this study was to determine the relationship between potential national factors such as life expectancy, average annual temperature, aging index, human development index, percentage of malnourished people in the population, extreme poverty, economic ability, health policy, population, and age distributions on the spread of COVID-19.

Here, we first applied segmented growth-curve models (logistic and Gompertz models) to the cumulative confirmed cases of 213 countries. Next, we applied the segmentation algorithm to divide the cumulative curve of COVID-19 cases into several segments of time series cases corresponding to a specific segment, which can then be modeled by the conventional growth curve models into a sigmoid curve. As the spread of COVID-19 has been prolonged, several countries experienced more than one wave (a wave implies a rising number of sick individuals, a defined peak, and then a decline [16] of the pandemic so that the cumulative confirmed cases could not be analyzed by conventional growth curve models, since cumulative cases cannot adopt the sigmoid curve. Thus, the study period of each country was divided into several segments—time periods corresponding to a specific wave using the segmentation algorithm (Appendix A) that systematically partitions COVID-19 cumulative confirmed cases for each country into several segments of times series cases corresponding to a specific wave, which can then be modeled by the conventional growth curve models into a sigmoid curve (for example, Figure 1).

The segmented growth curve models (GCMs) summarize the spread of COVID-19 into sets of three parameters α, β, and γ, where α is the maximum number of predicted cumulative confirmed cases, β is the time when we start to see a rise in the number of confirmed cases, and γ is the rate of spread of COVID-19. Thus, each country’s COVID-19 spread pattern was summarized into three GCM parameters. Then, a regression model was employed to investigate the relationship between 31 selected national factors from *Our World in Data* [17] and Korean Statistical Information Services (KOSIS) [18] such as life expectancy, average annual temperature, aging index, human development index, percentage of malnourished people in the population, and extreme poverty (Appendix A), and the spread of COVID-19, using the above parameters estimated from the segmented growth curve models. The significant relationship provides evidence that these variables may influence the spread of the novel SARS-CoV-2 virus across the globe.

## 2. Materials and Methods

### 2.1. ECDC COVID-19 Data

The COVID-19 data of daily confirmed cases and deaths can easily be downloaded from the European Centre for Disease Prevention and Control (ECDC) website [19,20,21]. ECDC is an EU agency aimed at strengthening Europe’s defenses against infectious diseases. Negative confirmed cases were corrected to 0, regarding it as abnormal data. Since data of cases on an international conveyance in Japan were included in a country list, we removed it. The data consisting of 213 countries from 1 January 2020 to 31 August 2020 were used in downstream analysis. 

Data smoothing was used to remove noise from a dataset, allowing important patterns to stand out. Thereafter, daily confirmed case data were smoothed by simple moving average (1) to reduce the effect of outliers and (2) to remove the weekly periodicity observed in the data. There were several outliers that showed greater or smaller abnormalities, which made it difficult to fit the statistical model. In addition, weekly periodicity was observed in the daily confirmed case data for many countries. Although we tried to present numerically through autocorrelation function, the trend had randomness, giving a limit to the analysis. Therefore, considering the period of 7 days, we set the window size to 7, and simple moving average (SMA) was used before model fitting as shown below
$$SMA=\frac{p_{M}+p_{M-1}+\ldots +p_{M-\left(n-1\right)}}{n}=\frac{1}{n}\overset{n-1}{\underset{i=0}{\sum }}p_{M-i},$$
where *p* is the number of confirmed cases.

### 2.2. National Factors

Time-independent national factor (Appendix A) datasets are publicly available datasets that are easily obtained from the *Our World in Data* website [22] and the Korean Statistical Information Services (KOSIS) [18]. The *Our World in Data* website provides data about research and data to make progress against the world’s largest problems such as poverty, disease, hunger, climate change, war, and existential risks. It mainly focuses on the large problems that continue to confront us for centuries or much longer, as well as the long-lasting forceful changes that gradually reshape our world. From this website, we obtained 15 time-independent social and economic factors assumed to be related to COVID-19 in the literature, such as population, population density, median age, being aged 65 or over, being aged 70 or over, GDP per capita, extreme poverty, cardiovascular death rate, diabetes prevalence, female smoker, male smoker, handwashing facilities, hospital beds per thousand people, life expectancy, and human development index [23,24,25,26,27,28,29].

The Korean Statistical Information Service (KOSIS) [18] website contains the national statistical database, which offers a full range of major domestic, international, and North Korean statistics produced by over 120 statistical agencies covering more than 500 subject matters as well as the latest data on international finance and economy from international organizations (i.e., IMF, World Bank, OECD). From the 26 variables, 13 were selected, which we assumed to be related to the spread of COVID-19. These variables were measured for several years. Therefore, we selected the year with the minimum number of missing values between 2016 and 2019, re-scaled by division with standard errors of the variables.

### 2.3. Analysis of the Spread of COVID-19 Using GCMs

Under this analysis, the growth curve models (GCMs) logistic model and Gompertz model were employed to model the transmission of COVID-19 using the cumulative confirmed cases for each country. These growth models are commonly used to explore risk factors and predict the probability of occurrence of a certain disease, investigate factors that control and affect growth, and extinction laws of the population [30]. The models take the following forms:

### 2.4. Logistic Model

 (1)$$Q_{t}=\frac{\alpha }{1+e^{\beta -\gamma \left(t-t_{0}\right)\;}}\;\;$$
where $Q_{t}$ is the cumulative confirmed cases, α is the maximum number of predicted cumulative confirmed cases, β is the time when we start to see a rise in the number of confirmed cases, γ is the increase rate of number of confirmed cases, *t* is the number of days since the first case occurrence, and $t_{0}$ is the time when the first case occurred.

### 2.5. Gompertz Model

 (2)$$\;Q_{t}=\alpha e^{-\beta e^{-\gamma \left(t-t_{0}\right)}}$$
where $Q_{t}$ is the cumulative confirmed cases, α is the maximum number of predicted cumulative confirmed cases, β is the time when we start to see a rise in the number of confirmed cases, γ is the increase rate of number of confirmed cases, *t* is the number of days since the first case, and $t_{0}$ is the time when the first case occurred.

### 2.6. Segmentation Algorithm

As the COVID-19 situation continues, fitting a growth curve model on daily confirmed cases over a long period of time has become impossible as it no longer takes on an s-curve (i.e. sigmoid function). To fit the above growth curve models, there is a need to divide the study period of countries experiencing more than one wave [31] (a wave implies a rising number of sick individuals, a defined peak, and then a decline) of the pandemic into several segments (the time during which cumulative confirmed cases follow the s-curve). Thus, we applied the segmentation algorithm, which can systematically divide study periods into several segments (or waves) for each country (Figure 1). 

Segmentation is a method of finding peaks and breakpoints, where a peak is the timestamp at which daily new confirmed case is highest in a segment, and breakpoint is the timestamp that splits the consecutive two segments in a time series dataset. To better see trends, we smoothed out the irregular roughness of the graph of daily confirmed cases. However, daily new confirmed cases have high randomness arising from (1) the fact that daily new confirmed cases have a periodicity of seven days (due to differences in daily new confirmed cases between weekends and weekdays) and (2) measure errors of one day. Therefore, we applied the Nadaraya–Watson kernel regression estimator (NWE) [32,33,34] with Gaussian kernel to smoothen the daily new confirmed cases as demonstrated in Figure 2 using South Korea’s daily confirmed cases as an example. For the convenience of notation, let $Y_{t}$ be the t-th daily new confirmed cases from data, and $\hat{f}\left(t\right)$ be the estimated t-th daily new confirmed cases using above NWE since 1 January 2020.

Peak detection (Algorithm 1; Appendix A) utilizes the first and second derivative test to find local maxima on convex function. $\hat{f}\left(t\right)$ has convexity when t is around peak due to the nature of epidemic dynamics. Considering daily new confirmed cases being discrete time series data, we found the location where the first difference is zero and second difference is negative (since $f\left(t\right)$ is not differentiable, we used difference operator instead of derivative): (3)$$\Delta \hat{f}\left(t\right)=0,\;\Delta^{2}\hat{f}\left(t\right)<0$$
where $\Delta \hat{f}\left(t\right)=\hat{f}\left(t+1\right)-\hat{f}\left(t\right)$ and $\Delta^{2}\hat{f}\left(t\right)=\Delta \hat{f}\left(t+1\right)-\Delta \hat{f}\left(t\right)$. 

For discontinuity and small variances of $\hat{f}\left(t\right)$, we used following condition:(4)$$\Delta \hat{f}\left(t\right)\Delta \hat{f}\left(t+1\right)\leq 0,\;\Delta^{2}\hat{f}\left(t\right)<-c\cdot {argmax}_{t\in \;T}|\Delta^{2}\hat{f}\left(t\right)|$$
where $c\in \left(0,\;1\right)$ is sensitivity level and T is the set of time indices from 1 January 2020 to 31 August 2020. In addition, 3 additional conditions ((a) exclusion of small peaks, (b) resolution criteria, and (c) exclusion of peaks that are vibrations on increasing trend) were used in peak detection to enhance robustness. After all the peaks were found, breakpoints (Algorithm 2; Appendix A) were selected either as timestamps that have the smallest daily new confirmed cases between two consecutive peaks or the timestamp where the cumulative confirmed case of the last segment saturates (that is, the last stage of the s-curve of last segment). Appendix A visualizes the segmentation process. Blue line represents the peak, and dotted sky-blue line represents breakpoint. In the first plot, the black solid line represents $\hat{f}\left(t\right)$ and the black dotted line represents $Y_{t}$. The second plot represents cumulative confirmed cases of $Y_{t}$(black dotted line), $\hat{f}\left(t\right)$(black solid line). The third and fourth plots are graphs of $\Delta \hat{f}\left(t\right),\;\Delta^{2}\hat{f}\left(t\right)$. In the fourth plot, the green dotted line represents sensitivity level. If $\Delta^{2}\hat{f}\left(t\right)$ is above the upper green dotted line, $\hat{f}\left(t\right)$ is concave. On the other hand, if $\Delta^{2}\hat{f}\left(t\right)$ is below the lower green dotted line, $\hat{f}\left(t\right)$ is convex. Within the third and fourth plots, Equation (4) can be validated. The segmentation algorithm was successfully applied to 134 countries from the 213 countries in the ECDC dataset, which met ${max}_{t\;\in \;T}\left(\hat{f}\left(t\right)\right)\geq 50$. If ${max}_{t\;\in \;T}\left(\hat{f}\left(t\right)\right)$ is too small, segmentation algorithm would be difficult to apply due to small variances in $\Delta^{2}\hat{f}\left(t\right)$. 

### 2.7. Segmented Growth Curve Models

Segmented growth curve models (segmented logistic model and segmented Gompertz model) fit the above-mentioned growth curve models ((1) and (2)) for each segment independently. These new models did not preserve continuity at breakpoints, but this did not matter since the objective of our analysis was to condense daily new confirmed cases into several parameters (α, β, γ) of the growth curves and not to accurately predict daily new confirmed cases. Equations (5) and (6) below are the segmented logistic and Gompertz models, respectively.
(5)$$Q\left(t\right)=\overset{n}{\underset{i=1}{\sum }}(\frac{\alpha_{i}}{1+e^{\beta_{i}-\gamma_{i}t}}+q_{i})I_{{Seg}_{i}}\left(t\right)$$
(6)$$Q\left(t\right)=\overset{n}{\underset{i=1}{\sum }}(\alpha_{i}e^{-\beta_{i}e^{-\gamma_{i}t}}+q_{i})I_{{Seg}_{i}}\left(t\right)$$
where, $q_{i}\left(i\geq 2\right)$ is the number of cumulative cases at ${\left(i-1\right)}^{th}$ breakpoint, and $I_{Seg_{i}}\left(t\right)$ is indicator function where $Seg_{i}$ is the set of indices of $i^{th}$ segment and $q_{1}=0$.

In this analysis, we only considered first and second segments, since most countries have 1 or 2 segments (1 segment: 62, 2 segments: 65, 3 segments: 7). The number of countries with three segments was very small, making the comparison analysis insignificant to use in the regression analysis. For countries with more than 2 segments, the analysis period was, therefore, cut off at the second breakpoint. For countries with 2 segments, segmented growth curve model then would produce two sets of parameters, one set from each segment. 

After the segmentation algorithm was applied to 134 countries, these countries were fitted to segmented logistic and Gompertz models. To filter out poorly fitted countries, we excluded countries whose MSSE (mean squared scaled error) was higher than 0.4, as defined below:(7)$$\mathrm{MSSE}=\frac{1}{N}\sum_{t=1}^{N}{\left(\frac{Y_{t}-\hat{Y_{t}}}{\bar{Y}}\right)}^{2}$$
where $Y_{t}$ is the daily new confirmed cases, $\hat{Y_{t}}$ is the predicted value for $Y_{t}$ by segmented logistic and Gompertz models, and $\bar{Y}$ is the mean of $Y_{t}$ for t = 1, …, N.

MSSE is a more suitable measure compared with MSE (mean squared error) or MAPE (mean absolute percentage error) because the MSE does not consider scales of population among each country, while MAPE overestimates its error when the number of daily new confirmed cases, and $Y_{t}$ is small. Among the 134 countries, 124 countries were fitted for the segmented logistic model and 119 countries for the segmented Gompertz model. Among the fitted countries, 5 countries were excluded due to failure of meeting the MSSE criteria of 0.4 for segmented logistic and segmented Gompertz models. Therefore, a total of 119 countries were used in the segmented logistic model, and 114 countries for the segmented Gompertz model (Figure 3). 

In addition, correlation analysis for segmented logistic and Gompertz models with the log-scaled of parameters was performed to determine the similarity between parameters of the two models (see Appendix A).

### 2.8. Regression Model

The above segmented growth curve models summarize the spread of the pandemic into three parameters ($\alpha_{1}$, $\beta_{1}$, $\gamma_{1}$) for countries with one segment, and into six parameters ($\alpha_{1}$, $\beta_{1}$, $\gamma_{1}$, $\alpha_{2}$, $\beta_{2}$, $\gamma_{2}$) for countries with two segments. Each of the parameters from the two segmented GCMs was regressed against the national factors shown in Figure 1 as follows: (8)$$y_{j}^{ik}=\theta_{0}^{ik}+\theta_{1}^{ik}x_{j}+\varepsilon$$
where $y_{j}^{ik}$ is one of the segmented GCM parameters (α, β, γ) for model i=1 (logistic), 2 (Gompertz), segment k=1, 2 and country j. $\theta_{0}$ and $\theta_{1}$ are regression coefficients, and $x_{j}$ is the national factor of country j. F-statistic was performed to test the significance of $\theta_{1}$ for each national factor to find out which variables had a significant relationship with *y*, a measure of the spread dynamics of COVID-19 for a country.

## 3. Results

### 3.1. Growth Curve Models Predicted the Spread of COVID-19 across Countries

In this analysis, we adapted and applied two GCMs: logistic and Gompertz models. Since the countries experienced more than one wave of the pandemic as of 31 August 2020, segmented GCMs were used to fit each wave independently, with each wave corresponding to a segment. Therefore, these models summarized the spread patterns of COVID-19 cumulative confirmed cases of 134 countries, i.e., three parameters ($\alpha_{1}$, $\beta_{1}$, $\gamma_{1}$) for countries with one wave (and therefore, one segment), and six parameters ($\alpha_{1}$, $\beta_{1}$, $\gamma_{1}$; $\alpha_{2}$, $\beta_{2}$, $\gamma_{2}$) for countries with two waves (two segments). Here, the differences between parameters estimated from logistic and Gompertz models among the countries are discussed (Appendix A). Figure 4 shows the differences between the parameter values estimated from GCMs among the countries. The *x*-axis represents the parameter related to the number of maximum predicted cumulative confirmed cases (α), while the *y*-axis represents the parameter related to the rate of spread of COVID-19 (γ).

Parameter estimation showed that the Philippines, India, and Brazil had the highest numbers of maximum predicted cumulative confirmed cases in the first segment of the pandemic using the logistic model (Figure 4A), while India and Zambia were shown to have the highest numbers using the Gompertz model (Figure 4B). In the second segment of the pandemic, the USA had the highest number of maximum cumulative confirmed cases using both GCMs (Figure 4C,D). Therefore, by 31 August 2020, the USA was the country with the greatest number of cumulative confirmed cases in the world. All the other remaining countries did not have notably large differences in their numbers of maximum predicted cumulative confirmed cases in both GCMs.

However, we observed somewhat large differences in the rate of spread of COVID-19 values among the countries. In the first segment of the pandemic, Djibouti, Malawi, and New Zealand had the highest rate of spread, while Sweden had the lowest rate of spread (γ) of COVID-19 among their populations (Figure 4A,B), according to both models. In the second segment, the Democratic Republic of Congo, Montenegro, and Cote d’Ivoire (Ivory Coast) had the highest rate of spread, while Iceland, Finland, the UK, Nepal, Australia, and Japan had the lowest rate of spread of COVID-19 in their populations, according to both models (Figure 4C,D).

Moreover, we observed that countries with the greatest numbers of predicted maximum cumulative confirmed cases had the smallest rate of spread and vice versa, in both models and segments (see Appendix A of log10α vs. γ). The correlation analysis (Pearson’s correlation) to determine the relationship between the parameters across the two models and segments (Appendix A) confirmed that the parameters had similar interpretation across models and segments, but a noticeable negative correlation (−0.5 and −0.55 for logistic, −0.66 and 0.7 for Gompertz) between α and γ parameters (Appendix A) was observed. This may explain the relationship observed between the numbers of maximum predicted cumulative confirmed cases and the rate of spread of COVID-19.

Furthermore, since the first day of the analysis period was set to the date when the number of cumulative confirmed cases exceeded 50 for each country, the population scale among countries was not considered. Thus, the time when we started to see a rise in the number of confirmed cases (β) did not produce consistent results between segments and models as the other parameters did, although its interpretation was the same between the models. Therefore, its results and any analysis concerning it were not a focus in our study, and its results were relegated to the Appendix A for those interested. In addition, β showed minimal correlation (−0.089 for logistic and 0.15 for Gompertz) between the two segments and with other parameters (e.g., −0.069 and 0.19 for logistic, 0.077 and 0.10 for Gompertz) in the same model, but it showed a strong positive correlation between the models (0.88 and 0.95).

### 3.2. The Relationship between National Factors and the Spread of COVID-19

Regression model was employed to investigate the relationship of selected national factors (Appendix A) reasonably assumed to be related to COVID-19 and the spread of COVID-19 using the number of maximum predicted cumulative confirmed cases, α, and the rate of spread of the pandemic, γ, estimated from the segmented GCMs. The 31 national factors included developmental (called World Development Indicators by World Bank [35]) and non-developmental variables related to population, age distribution, health, and environment (Appendix A).

The objective of our analysis was to determine whether these factors influence the spread of COVID-19. From the segments in each growth curve model, our focus was on whether (1) the differences in the size of the estimated coefficients and (2) the estimated coefficients were statistically significant between two models and two segments. We used a 5% significance level in this analysis. Statistically significant results provided evidence for the possibility of these factors influencing the spread of COVID-19.

For the number of maximum predicted cumulative confirmed cases (α), several national factors turned out to be significant, such as population, annual precipitation, pharmaceutical sales, and imports to GDP ratio (Figure 5A,C). However, population was the only variable that was outstandingly significant in both segments (1, 2) and models (logistic, Gompertz). The rate of spread of COVID-19 (γ) was significantly related to 19 national factors. For example, age-related variables such as aging index, share of population aged 65 and older, share of population aged 70 and older, median age and life expectancy, health-related variables such as life cardiovascular death rate, share of female and male smokers in the population and percentage of malnourished people in the population, hospital beds per thousand, extreme poverty and human development index, cultural variables such as international travelers from a country and number of foreign visitors to a country, and environmental factors such as average annual temperature (Figure 5B,D).

In addition, a relationship between the size of coefficient values (of the relationship between national factor and GCM parameter) and significance of national factors was observed, whereby significant variables generally had larger coefficient values than non-significant variables (Figure 6, Appendix A). Our results provide evidence of the influence of these significant national factors such as population, aging index, median age, cardiovascular death rate, extreme poverty, annual precipitation, number of foreign visitors and international travelers, on the spread of COVID-19 across the globe. Moreover, we rarely observed a change in signs of the coefficients of the significant variables between models.

The number of maximum predicted cumulative confirmed cases is significantly influenced by only population in both the two GCMs and segments of each model (Figure 7B). The countries with the highest value of maximum predicted cumulative confirmed cases (India, the USA, Brazil, the Philippines, and Zambia) had the highest population sizes in the world as of 31 August 2020 [36]. In addition, the USA, India, and Brazil have been, in that order, the countries hardest hit by the COVID-19 pandemic worldwide [37,38], showing a relationship between population sizes and the number of confirmed cases. i.e., the spread of COVID-19. High population may bring about congestion of people and higher rate of person-to-person contacts among the people in public places. However, other population dynamic factors may bring about this observation.

The rate of spread of COVID-19 is influenced by 16 significant variables in the Gompertz model, and 10 significant variables in the logistic model (Figure 7A). Age-related variables. i.e., aging index, median age, percentage of the population aged 65 or 70 and older, and life expectancy are significant in both models and segments. Aging is linked mainly with deteriorating immune system [39] and other common conditions such as hearing loss, cataracts and refractive errors, back and neck pain and osteoarthritis, chronic obstructive pulmonary disease, diabetes, depression, and dementia, wherein several of these conditions can be experienced at the same time [40,41]. The risk for severe illness with COVID-19 increases with age, with older adults being at a greater risk of requiring hospitalization and dying of COVID-19 when diagnosed in comparison with younger people. This is due to already deteriorating immune system, pre-existing conditions, and underlying medical problems (cardiovascular disease, diabetes, chronic respiratory disease, and cancer) that also makes them prone to newer infections [22,42,43,44]. This includes other variables such as cardiovascular death rate [45] and the percentage of female and male smokers in the population.

One in five (20%) adults in the world smoke tobacco [46], being one of the world’s largest health problems. Active smoking and a history of smoking (cigarettes, waterpipes, bidis, cigars, heated tobacco products) may lead an individual to being vulnerable to contracting COVID-19, having been linked to increased severity of COVID-19 illness due to the health complications, wrecking mainly the immune system, especially on the lungs (epithelial cells), which is a primary site of target of SARS-CoV-2 [47,48,49]. Moreover, the act of smoking involves contact of fingers (and possibly contaminated cigarettes) with the lips, which increases the possibility of transmission of viruses from hand to mouth. Smoking waterpipes, also known as shisha or hookah, often involves the sharing of mouth pieces and hoses, which could facilitate the transmission of the COVID-19 virus in communal and social settings [48]. It is reported that Montenegro has 46% smoking prevalence, being a country with the second highest rate of spread of COVID-19 in the second segment of analysis [46], while OECD member countries were found to have a prevalence of 23.50% as of 2016. African countries have some of the lowest levels of smoking in the world [46].

Extreme poverty impairs rapid response of the government to newer pandemics or even other disasters, leaving its people highly susceptible to the infections. It influences a government’s preparedness to deal with disasters (new pandemics included) and interferes with health system response such as drugs, protective gear, information campaign, and the inability of poor health systems to handle newer pandemics. Malnutrition increases one’s susceptibility to and severity of infections and is thus a major component of illness and death from disease. The risk of death is directly correlated with the degree of malnutrition [50,51,52]. Malnutrition is consequently the most important risk factor for the burden of disease in developing countries. Malnutrition continues to be a major public health problem throughout the developing world, particularly in southern Asia and sub-Saharan Africa [53,54]. 

Number of international travelers and foreign visitors increases the chance of spreading and catching the SARS-CoV-2 virus among the population [55], mainly due to importation and exportation of cases, leading to many domestic travel restrictions and flight suspensions between countries [39,56]. Accelerated by human migration, exported COVID-19 cases have been reported in various regions of the world, including Europe, Asia, North America, and Oceania [57]. National competitiveness that covers areas such as economic performance, government efficiency, corporate efficiency, and infrastructure, influencing the rate of spread of COVID-19, may involve all the above-mentioned areas, for example, government efficiency in the response to disaster may determine the overall outcome of the situation. South Korea’s response to COVID-19, especially in the early stages of the pandemic, has been widely praised and encouraged to be emulated around the globe, showing the importance of national competitiveness in response to COVID-19 [58,59]. Although climate factors may have influenced the rate of spread of COVID-19, they may have had a smaller effect size compared to the other significant factors. As a result, climate factors did not turn out to be consistently significant across models and segments (only 1 model, 1 segment). A recent review has addressed the role of climate change in the emergence and re-emergence of infectious diseases worldwide, indicating that temperature is an important environmental condition determining the success of infectious agents [60,61].

## 4. Discussion

In this study, we investigated the relationship of 31 national factors from KOSIS and *Our World in Data* on the spread of COVID-19 in 134 countries. First, we modeled the spread of COVID-19 using segmented logistic and Gompertz models, and then we investigated the influence of national factors on the spread of COVID-19. We observed that some factors were significant in both GCMs or the two segments for each model, while others were significant in only one model or segment, which implies a change in segments. We believe that although the curves from GCMs can describe similar behavior in some phases of growth, with one of the most important differences being that the Gompertz process is asymmetric, whereas the logistic curve is a symmetric process, explaining the differences observed in the results of the two models. Therefore, using a given growth curve model can have a substantial impact on forecasting [36]. By building two models and analyzing the results (Figure 7), we concluded that our findings provide reasonable proof that the significant variables influence the spread of COVID-19.

We observed that the number of maximum predicted cumulative confirmed cases was significantly influenced by only one factor, while the rate of spread of COVID-19 was influenced by seven factors, in both the two GCMs and segments of each model (Figure 5). This made the rate of spread of the pandemic the most influenced aspect of the spread of COVID-19 among countries among the two parameters. Moreover, we found out that the number of maximum predicted cumulative confirmed cases (α) did not vary much across countries (although we observed a few outliers, e.g., the USA, India, Brazil, the Philippines, and Zambia), while the rate of spread of COVID-19 (γ) varied greatly across countries. We observed that α was only mainly influenced by population (Figure 7B), while γ was significantly influenced by many variables (Figure 7A). This may explain the differences observed in the rate of spread of COVID-19 among countries in comparison with the number of maximum predicted cumulative confirmed cases. It was seen that different variables influenced the spread of COVID-19 at different segments of the pandemic. 

We saw the influence of population size on the spread of COVID-19. Among the hardest hit countries by the COVID-19 pandemic in the world, the USA, India, and Brazil are also among the countries with the largest populations in the world. Some countries with the highest number of maximum predicted cumulative confirmed cases (Zambia, India, Brazil, and the Philippines) and the highest rate of spread of COVID-19 (Democratic Republic of Congo and Malawi) have a large percentage of their population living in extreme poverty [62,63] and in a malnourished state [64,65], as well as having the youngest populations (especially African countries) in the world [66]. Moreover, in the first segment, Iceland, South Korea, China, New Zealand, and Australia, which had a high rate of spread of COVID-19, are characterized as having older populations, longer life expectancy, higher GDP per capita, higher cardiovascular death rate, large percentage of population that smoke daily [67], better health systems, and little to no malnutrition [66,68,69,70,71]. Clearly, we observed the influence of these variables on the spread of COVID-19 [35]. However, most of these countries, in addition to Japan, the UK, Italy, Germany, and the United Arabs Emirates (despite having the characteristics listed above), also had the lowest rate of spread of COVID-19 in the second segment of the pandemic (Figure 4). This could have been due to the influence of government-implemented policies such as “lockdowns” in response to the spread of COVID-19.

However, there are some limitations in our analysis. For example, a key limitation of this analysis is that although we modeled the spread of COVID-19 for 134 countries, the GCMs still produced some missing parameter values (10 countries in the logistic model, 14 countries in the Gompertz model) between the segments and models for some countries mainly due to failure of convergence (Appendix A), which may have affected comparison and therefore the interpretation of the results. Moreover, we could only fit the model up to August 31, 2020 because beyond that, more than two segments would have to be modeled as currently many countries are experiencing their third wave or beginning their fourth wave of the pandemic, which was challenging to the segmentation algorithm. In the future, we hope to improve on this algorithm and then be able to study the other waves of the pandemic and solve the problem of failure of convergence in the models. While the relationship between several national factors and COVID-19 via regression has been studied at the univariate level, multivariate analysis/regression has not been performed to adjust for the influence of one factor on the association between another factor and COVID-19. We hope to perform this type of analysis too by including some sort of variable/feature selection, and then study the relationship between the selected feature set of national factors and COVID-19 at multivariate level.

Moreover, COVID-19, which is a contact-transmissible infectious disease and is said to spread through the population via direct contact between individuals [2,72,73] as the main route of transmission, elicited a wide range of control measures from each country, aimed at reducing the amount of mixing in the population [74,75]. These government-implemented policies have already been shown to mitigate and suppress the pandemic [76,77]. It was determined that highly effective contact tracing and case isolation is enough to control a new outbreak of COVID-19 within three months in most scenarios [78]. However, it was very important to include these policies or to model their effects in our analysis, since these policies may have influenced the results observed from the segmented GCMs. However, our analysis could not, since our approach cannot handle time-dependent variables such as the containment policies. Therefore, we could not control for this bias in our analysis, as some may argue on this topic. In the future, we hope to consider the impact of government-implemented policies on the spread of COVID-19 in our analysis using other models. 

In addition, the role of host genetics interaction and COVID-19 progression has gained a large amount of interest as one of the factors being proposed to influence the spread of COVID-19 [79]. For example, the difference in terms of incidence of COVID-19 observed between the northern and southern regions of Italy was attributed to genetics as being one of the factors causing this inhomogeneous distribution of cases [24]. However, we analyzed a country’s COVID-19 pandemic situation instead of specific COVID-19-confirmed individuals. This made it difficult to include genetic information in the current models. However, provided that ethnic or ancestral difference data of each of the countries analyzed is available, we can indirectly analyze effects of genetics using the ethnical differences of a given country as another covariate in the GCMs. We hope to model the role of genetics in relation to the spread of COVID-19 in a future study. 

Furthermore, we also hope to repeat this analysis using number of cumulative COVID-19 death cases. The number of death cases are just as important as confirmed cases in the understanding of influential factors and epidemiological characteristics of COVID-19, as we believe that COVID-19 death cases will provide more insight as they may be more related to age distributions and health-related variables.

## 5. Conclusions

Much is still unknown about the clinical and epidemiological characteristics of COVID-19, such as individual risk factors for contracting the virus and infections from asymptotic cases. However, from the above discussions, our findings show the relationship between age distributions, life expectancy, malnutrition, extreme poverty, cardiovascular death rate, smoking, and population size and the spread of COVID-19. We hope these studies will provide important information for policymakers and governments in making informed scientific decisions while considering a country’s economy, population dynamics, climate, and health system, which would likely have the most impact in future prevention works against similar infectious diseases.

## Figures and Tables

**Figure 1 ijerph-18-07592-f001:**
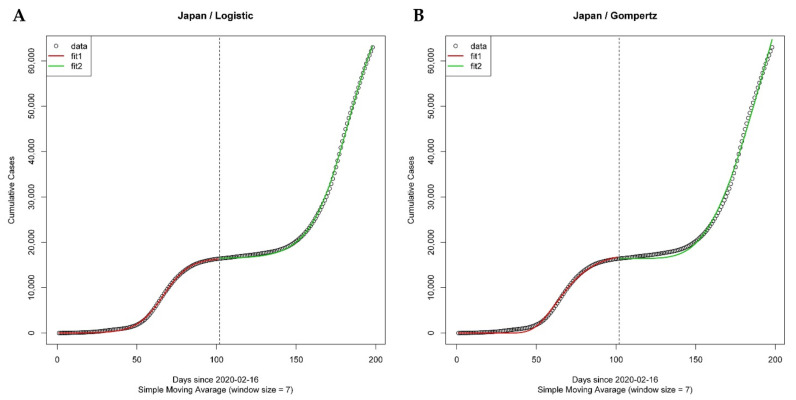
Cumulative confirmed cases divided into two segments using the segmentation algorithm. Japan is the typical country with two waves (red: first segment and green: second segment). (**A**) Epidemic segmented growth curve of COVID-19 fitted by the logistic model. Estimated parameters were $(\alpha_{1},\;\beta_{1},\;\gamma_{1},\;\alpha_{2},\;\beta_{2},\;\gamma_{2}$) = (16549.7588, 7.8244, 0.1231, 58877.8030, 6.7353, and 0.0859, respectively). (**B**) Epidemic segmented growth curve of COVID-19 fitted by the Gompertz model. Estimated parameters were $(\alpha_{1},\;\beta_{1},\;\gamma_{1},\;\alpha_{2},\;\beta_{2},\;\gamma_{2}$) = (17,531.5024, 89.4251, 0.0760, 98,622.3568, 15.6933, and 0.0325, respectively).

**Figure 2 ijerph-18-07592-f002:**
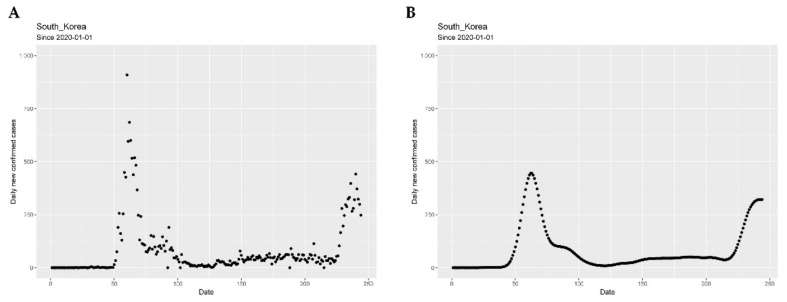
Daily new confirmed cases before (**A**) and after smoothing using Nadaraya–Watson kernel regression (**B**).

**Figure 3 ijerph-18-07592-f003:**
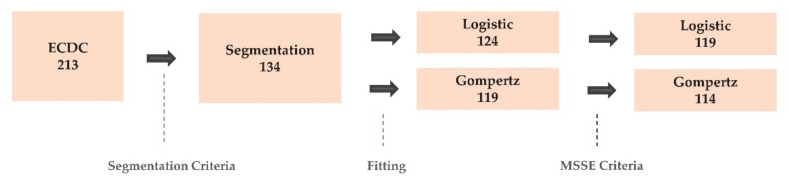
Differences in the number of countries across segmentation, growth curve models, and MSSE criteria. For the segmented logistic model, 124 countries were fitted, and for the segmented Gompertz model, 119 countries were fitted. To check (validate) the goodness of fit of the above 2 models, we employed MSSE (mean squared scaled error) criteria. For each of the two models, 5 countries showed high MSSE, and therefore those 5 countries were excluded in the subsequent analysis (Aruba, Equatorial Guinea, Krygyzstan, Rwanda, and Thailand for the segmented logistic model, and China, Equatorial Guinea, Kyrgyzstan, Rwanda, and Zambia for the segmented Gompertz model).

**Figure 4 ijerph-18-07592-f004:**
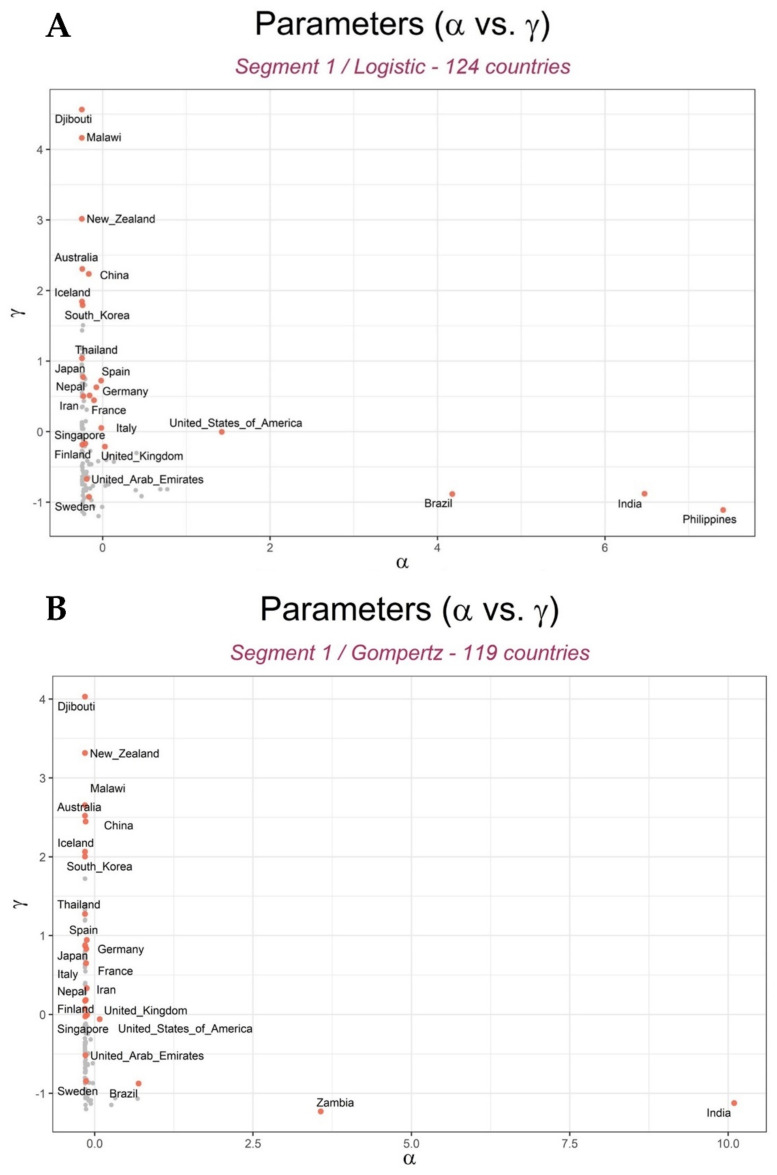
Differences in values of maximum predicted cumulative cases (α) and rate of spread of COVID-19 observed (γ) among countries. (**A**,**C**) The variation of α vs. γ using the logistic model for the first and second segments, respectively. (**B**,**D**) The variation of α vs. γ using the Gompertz model for the first and second segments, respectively.

**Figure 5 ijerph-18-07592-f005:**
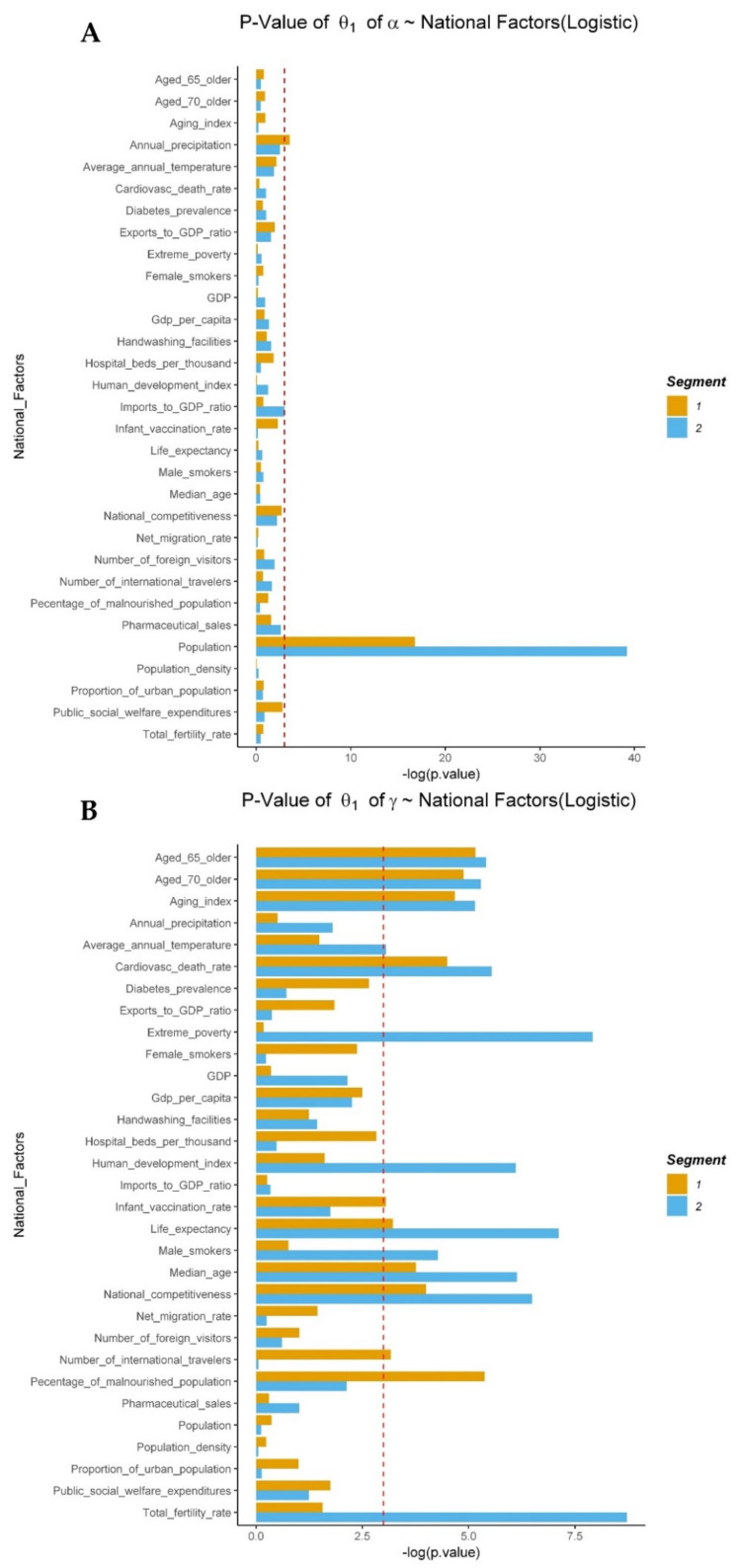
*p*-values of coefficients of national factors with α and γ. Population, annual precipitation, pharmaceutical sales, and imports to GDP ratio were statistically significant with α, the number of maximum predicted confirmed cases (**A**,**C**). Age-related factors, population, percentage of malnourished population, life expectancy, temperature, etc. were significantly related with γ, rate of spread of COVID-19 (**B**,**D**) (see Appendix A for the relationship of national factors with β).

**Figure 6 ijerph-18-07592-f006:**
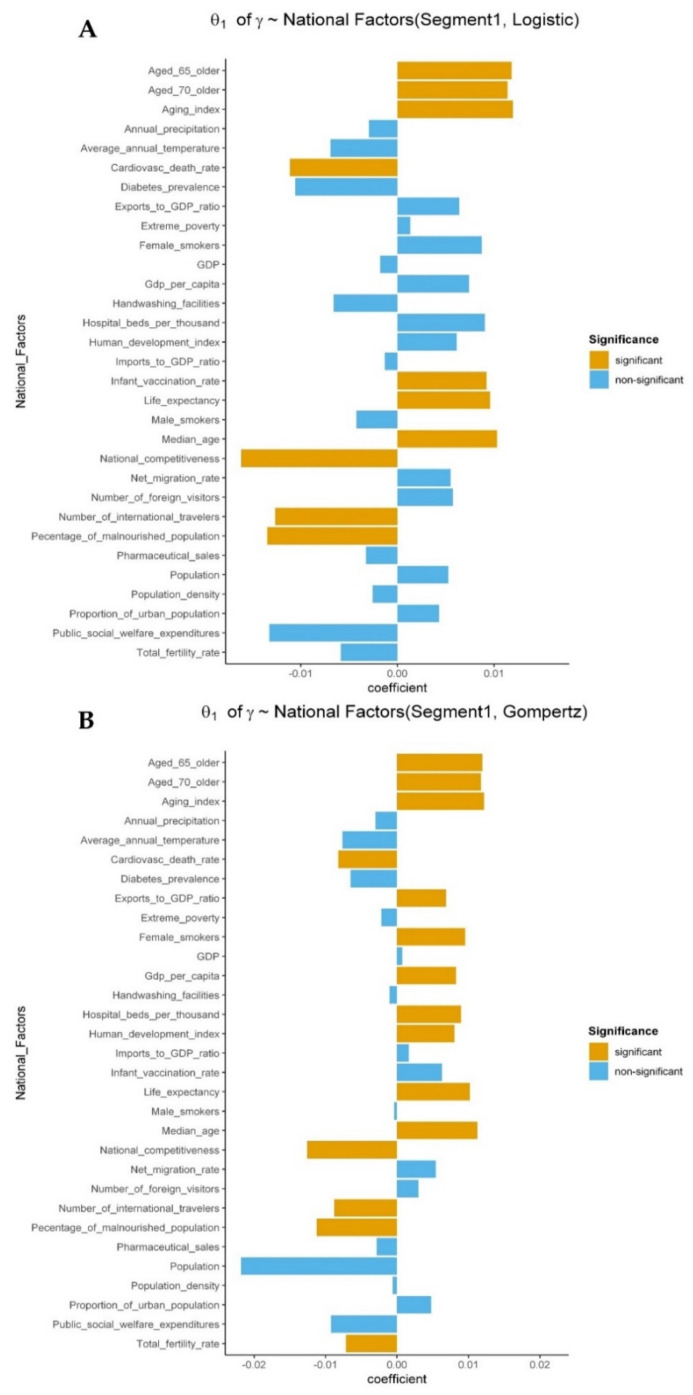
Coefficients of the relationship between national factors and γ. $\theta_{1}$ is the coefficient of the relationship between a national factor and a GCM parameter. Significant national factors (orange) had large coefficients compared with non-significant factors (blue) in both Gompertz (**A**,**C**) and logistic models (**B**,**D**).

**Figure 7 ijerph-18-07592-f007:**
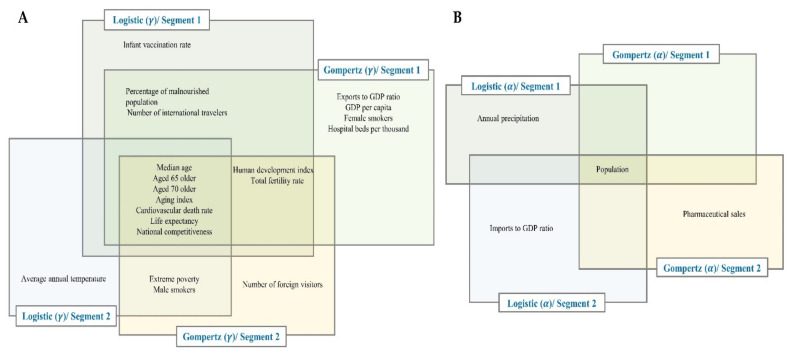
Significant national factors with number of maximum predicted cumulative confirmed cases (α) and rate of spread of COVID-19 (γ). Median age, being aged 65 or older, being aged 70 or older, aging index, cardiovascular death rate, life expectancy, and national competitiveness were the only national factors that were found to be significant across the two models and segments (**A**). Population was significant across the two models and segments (**B**) (see Appendix A for significant national factors with β).

## Data Availability

Publicly available datasets were analyzed in this study. These datasets can be found at the data links provided in the references. All data that were required to evaluate the conclusions in the paper are present in the paper and/or the Appendix A. All additional data used are available from the authors.

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
