# Peer review of "Which National Factors Are Most Influential in the Spread of COVID-19?"

_ijerph, 2021, doi:10.3390/ijerph18147592_

Round 1

Reviewer 1 Report

The authors in this manuscript explains their analysis of some key time independent national factors that influence the spread of COVID 19.

Here are my comments,

The manuscript is properly organized and well written. However the figures 5 & 6 on page 10 & 11 are not visible. Please include enlarged pictures.

Given the originality and broader interest in the scientific  community I recommend the editors to accept this manuscript for publication with minor changes.

About the 2 - Figures that are not at all visible

Author Response

We thank the reviewer for pointing out this issue. In response to the reviewer’s comment, we produced Figure 5 and Figure 6 with better resolutions in the revised manuscript. 

Reviewer 2 Report

To the best of my knowledge, I believe the manuscript is of very good quality and needs no major changes, since it is well structured, clear and well organized, conceptually and methodologically. There are some formal changes needed, as I mentioned, namely the figures 5 and 6 on page 10-11 are not visible. Pictures should be bigger and have better resolution.

The article and research is very well designed and provides very interesting results.

Author Response

We thank the reviewer for this kind comment and we appreciate it greatly. In response to the reviewer’s comment on the figures, we changed Figure 5 and Figure 6 on pages 10-11 to figures with better resolutions, in the revised manuscript.

Reviewer 3 Report

I read with interest the article entitled "Which national factors are most influential in the spread of COVID-19?”. The article is interesting and well written, but the authors need to expand on some of the aspects listed below.

Major Concerns

  • From the point of view of the mathematical model, there are few notes to be made. The descriptive part should instead be expanded by talking not only about health policies in general but also about the response of doctors to the pandemic. (preparation of the health system Nioi, Matteo, et al. "COVID-19 and Italian healthcare workers from the initial sacrifice to the mRNA vaccine: Pandemic chrono-history, epidemiological data, ethical dilemmas, and future challenges." Frontiers in Public Health 8 (2020).; d'Aloja, Ernesto, et al. "COVID-19 and medical liability: Italy denies the shield to its heroes." EClinicalMedicine 25 (2020).; : Matteo Nioi, et al. Fear of the COVID-19 and medical liability. Insights from a series of 130 consecutives medico-legal claims evaluated in a single nstitution during SARS-CoV-2-related pandemic. Signa Vitae. 2021. doi:10.22514/sv.2021.098.
  • Still in the discussion, factors such as genetics, climate, health policies but also restrictions in general should be discussed more broadly. [Napoli, P. E., Nioi, M., & Fossarello, M. (2021). The “Quarantine Dry Eye”: The Lockdown for Coronavirus Disease 2019 and Its Implications for Ocular Surface Health. Risk management and healthcare policy, 14, 1629.]

Minor Concerns

  1. The quality of the figures should be improved.
  2. Given the vastness of the topic, the number of citations is insufficient.

Author Response

We thank the reviewer for pointing out these issues and for suggesting many relevant references. After reviewing the references suggested, we included some of them in the list of our references. Also, we looked up more other references given the vastness of this topic. In addition, all figures were revised to better resolution in the manuscript.

Round 2

Reviewer 3 Report

I read the revised version of the paper entitled "Which National Factors are Most Influential in the Spread of COVID-19? ". I appreciated the changes made but I don't think the paper is ready for publication yet.

Considering the subject dealt with, I consider the paper incomplete without a hint of genetics. Consolidated literature is giving more and more weight to this factor (for example in countries where malaria has been endemic). It is essential that the authors dedicate a sub-chapter to the theme.

Although I liked the climate report, quote 40 talks about "global warming". Authors are asked for a more appropriate quote that speaks of the change in diffusion resulting from normal seasonal variations.

Although the list of  references has been expanded, it is requested to reconsider what was suggested in the previous round..

I believe that authors should meet these concerns to make the article suitable for publication.

Author Response

We thank the reviewer for the detailed recommendations and comments. These are our responses to the reviewer. Red letters are revised text in the main manuscript.

I read the revised version of the paper entitled "Which National Factors are Most Influential in the Spread of COVID-19? ". I appreciated the changes made but I don't think the paper is ready for publication yet.

Considering the subject dealt with, I consider the paper incomplete without a hint of genetics. Consolidated literature is giving more and more weight to this factor (for example in countries where malaria has been endemic). It is essential that the authors dedicate a sub-chapter to the theme.

à We thank the reviewer for pointing out this issue. In this study, we analyzed a country’s COVID-19 pandemic situation instead of specific COVID-19 confirmed individuals. This makes it difficult to use genetic information in the current models. But, provided data on the ethnic or ancestral differences of each of the country’s analyzed is available, effects of genetics can be analyzed indirectly using the information on ethnic differences, in our growth curve models. We dedicated a paragraph in our Discussion addressing this as a future study goal.

In addition, the role of host genetics interaction and COVID-19 progression has gained a lot of interest as one of the factors being proposed to influence the spread of COVID-19 (82). For example; the difference in terms of incidence of COVID-19 observed between the Northern and the Southern regions of Italy was attributed to genetics as one of the factors causing this inhomogeneous distribution of cases [73]. However, we analyzed a country’s COVID-19 pandemic situation instead of specific COVID-19 confirmed individuals. This made it difficult to include genetic information in the current models. However, provided ethnical or ancestral differences data of each of the country’s analyzed is available, effects of genetics can be analyzed indirectly using the ethnical differences of a given country as another covariate in the GCMs. We hope to model the role of genetics in relation with the spread of COVID-19 in our future study.

Although I liked the climate report, quote 40 talks about "global warming". Authors are asked for a more appropriate quote that speaks of the change in diffusion resulting from normal seasonal variations.

à We thank the reviewer for pointing out this issue. Following the reviewer’s comment, an additional reference that addresses the issue of seasonal variations was added.

Rayan R. A. (2021). Seasonal variation and COVID-19 infection pattern: A gap from evidence to reality. Current opinion in environmental science & health, 20, 100238. https://doi.org/10.1016/j.coesh.2021.100238

Although the list of references has been expanded, it is requested to reconsider what was suggested in the previous round.

à We thank the reviewer for pointing out this issue again. The following four references have been suggested by the reviewer. In the former revision, we added two references to our manuscript.

[Nioi, Matteo, et al. "COVID-19 and Italian healthcare workers from the initial sacrifice to the mRNA vaccine: Pandemic chrono-history, epidemiological data, ethical dilemmas, and future challenges." Frontiers in Public Health 8 (2020)].

In this current revision, we considered two more references;

[d'Aloja, Ernesto, et al. "COVID-19 and medical liability: Italy denies the shield to its heroes." EClinicalMedicine 25 (2020)].

[Matteo Nioi, et al. Fear of the COVID-19 and medical liability. Insights from a series of 130 consecutives medico-legal claims evaluated in a single institution during SARS-CoV-2-related pandemic. Signa Vitae. 2021. doi:10.22514/sv.2021.098].

However, the below reference was not in line with the topic of this paper so it wasn’t added to our list of references.

[Napoli, P. E., Nioi, M., & Fossarello, M. (2021). The “Quarantine Dry Eye”: The Lockdown for Coronavirus Disease 2019 and Its Implications for Ocular Surface Health. Risk management and healthcare policy, 14, 1629.]

I believe that authors should meet these concerns to make the article suitable for publication.

à We thank the reviewer for pointing out several important issues, which resulted in a great improvement of the revised manuscript. We now believe that the revision is ready for publication.